# Early Consumption of Cannabinoids: From Adult Neurogenesis to Behavior

**DOI:** 10.3390/ijms22147450

**Published:** 2021-07-12

**Authors:** Citlalli Netzahualcoyotzi, Luis Miguel Rodríguez-Serrano, María Elena Chávez-Hernández, Mario Humberto Buenrostro-Jáuregui

**Affiliations:** 1Laboratorio de Neurociencias, Departamento de Psicología, Universidad Iberoamericana Ciudad de México, Prolongación Paseo de la Reforma 880, Lomas de Santa Fé, Ciudad de México 01219, Mexico; cnetza@gmail.com (C.N.); cosmonauta84@yahoo.com.mx (L.M.R.-S.); mariele_chavez@yahoo.com (M.E.C.-H.); 2Centro de Investigación en Ciencias de la Salud (CICSA), FCS, Universidad Anáhuac México Campus Norte, Huixquilucan 52786, Mexico; 3Laboratorio de Neurobiología de la alimentación, Facultad de Estudios Superiores Iztacala, Universidad Nacional Autónoma de México, Tlalnepantla 54090, Mexico

**Keywords:** cannabinoids, endocannabinoid system, adult neurogenesis, behavior, memory, learning, stress, anxiety

## Abstract

The endocannabinoid system (ECS) is a crucial modulatory system in which interest has been increasing, particularly regarding the regulation of behavior and neuroplasticity. The adolescent–young adulthood phase of development comprises a critical period in the maturation of the nervous system and the ECS. Neurogenesis occurs in discrete regions of the adult brain, and this process is linked to the modulation of some behaviors. Since marijuana (cannabis) is the most consumed illegal drug globally and the highest consumption rate is observed during adolescence, it is of particular importance to understand the effects of ECS modulation in these early stages of adulthood. Thus, in this article, we sought to summarize recent evidence demonstrating the role of the ECS and exogenous cannabinoid consumption in the adolescent–young adulthood period; elucidate the effects of exogenous cannabinoid consumption on adult neurogenesis; and describe some essential and adaptive behaviors, such as stress, anxiety, learning, and memory. The data summarized in this work highlight the relevance of maintaining balance in the endocannabinoid modulatory system in the early and adult stages of life. Any ECS disturbance may induce significant modifications in the genesis of new neurons and may consequently modify behavioral outcomes.

## 1. A Worldwide View of Cannabinoid Consumption

Drug use worldwide is an important public health issue since the number of people that use legal and illegal drugs has been increasing. As an example, alcohol remains the most widely used substance of abuse in the world. In the global status report on alcohol and health, 2018 edition, the World Health Organization (WHO) stated that 43% of the world population aged 15 years or over have consumed alcohol in the past 12 months (2016), indicating that 2.3 billion people are current drinkers [1]. Regarding tobacco consumption, the United Nations Organization reported that 23.2% of adults globally were recurrent smokers [2]. Additionally, marijuana (*Cannabis* sp.) is still the most commonly used illegal drug. Worldwide, the number of cannabis users in 2018 was estimated to be 192 million, corresponding to a prevalence of 3.86%. North America has the highest consumption rate at 14.56%, followed by Australia and New Zealand (10.6%) and West and Central Africa (9.3%) [3]. In the United States of America (USA), marijuana use has been consistently increasing since 2007, particularly among young adults from 18–25 years old, but also in adults older than 25 years [3]. A meta-analysis using wastewater-based epidemiology (an alternative for human biomonitoring and a promising tool with which to estimate drug consumption in the population) recently reported that cannabis had the highest consumption rate (7417.9 mg/day/1000 people), followed by illicit drugs such as cocaine, morphine, methamphetamine, codeine, methadone, ecstasy, amphetamine, and methadone [4]. Table 1 summarizes the key statistics regarding drug abuse consumption in users aged 15 years old and above.

Although other countries, such as the USA, legalized the medicinal and controlled consumption of some derivatives of cannabis in the 1990s, in the 2000s, the approval of its recreational use began. In recent years, several countries in the Americas, including Uruguay (2013), Canada (2018), and up to 17 states (such as Oregon, Washington, and California), two territories, and the District of Columbia in the USA [5], have legalized the recreational use of marijuana. It is essential to understand the effect that this legalization has had on the use of cannabis in the general population, and young people in particular.

In December 2013, Uruguay became the first country in the world to legalize the sale, cultivation, and distribution of recreational cannabis [6]. A survey conducted in 2018 reported that 8.9% of the population aged 15–65 years used marijuana in the month before the survey. However, cannabis use prevalence in young people aged 19–25 years increased up to 20.8%, followed by a prevalence of up to 16.4% among those aged 26–35 years [7]. A recent study by Laqueur et al. (2020) estimated the impact of this legalization on adolescents, perceived availability, and perceived risk of marijuana use. Researchers have found no evidence of a legalization effect on cannabis use or the perceived risk of use compared with prelegalization data. Nevertheless, an increase in student perception of marijuana availability (58% observed vs. 51% synthetic control) following legalization was reported. The authors conclude that the noncommercial model of national cannabis legalization may not lead to an increase in adolescent marijuana use in the short term [8].

Rotermann (2019) reported that in Canada, from 2004 to 2017, cannabis use decreased among 15- to 17-year-olds, remained stable in people 18 to 24 years old, and increased among adults aged 25 to 64 years. The Canadian federal government legalized nonmedical cannabis use by adults in October 2018. From 2018 to 2019, an increase from 14% to 18% was observed in the rates of marijuana use; mainly, the rate of marijuana use in males increased from 16% to 22%. However, marijuana use rates for females (13%) and seniors (4%) remained mostly stable. The same study reported that in 2019, approximately 60% of regular consumers reported using at least one cannabis product. On average, 27.5 g of dried marijuana was consumed by each user over three months [9].

Rusby et al. (2018) proposed that there may be an immediate impact of legalization of recreational marijuana in Oregon (USA), increasing its use in youths (13–15 years old) who had already started using the drug [10]. However, legalization did not increase cannabis use in age-matched people who did not use it previously. These data became vital since it is well known that early use of marijuana results in a 2 to 5-fold greater probability of abuse and/or dependence on other illicit drugs (such as cocaine, methamphetamine, and heroin) compared with that in subjects who had not been exposed to cannabis [11]. Marijuana intake usually starts during late adolescence or early adulthood, at approximately 15–24 years old, and drastically decreases in adulthood [11]. Therefore, the data presented by Rusby et al. (2018) are of particular importance.

It is essential to know more about the major components of cannabis, particularly its effects and the mechanisms by which it acts on the nervous system. Additionally, it is clear that the legalization of marijuana in different countries makes this task even more important. As previously mentioned, the highest consumption rate is in the young population, particularly adolescents, so focusing on studies on this stage of life is of particular importance.

## 2. The Endocannabinoid System (ECS)

In the 1960s, the psychoactive components of cannabis (cannabinoids, among others) were isolated for the first time [12,13], and their structure was analyzed [14]. Since then, hundreds more compounds, including tens of cannabinoids, have been isolated [15,16]. Among them are cannabidiol (CBD), which lacks psychotropic effects, and Δ9-tetrahydrocannabinol (Δ9-THC), which mediates the euphoric effects of this drug by binding to cannabinoid receptors in the central nervous system (CNS) [17,18].

The ECS originally comprises two cannabinoid receptors (type 1: CB1, and type 2: CB2), endogenous ligands or endocannabinoids named anandamide (AEA) and 2-arachidonyl-glycerol (2-AG), and proteins responsible for the biosynthesis, transport and degradation of endocannabinoids (see Figure 1 for a schematic representation of cannabinoid function at the synapse and Table 2 for a summary of the effect of different compounds in the ECS). Some of the main enzymes that have been identified to be involved in endocannabinoid biosynthesis and degradation are (1) N-acylphosphatidylethanolamine-specific phospholipase D (NAPE-PLD), which catalyzes the synthesis of AEA; (2) fatty acid amide hydrolase (FAAH), which catalyzes the hydrolysis of AEA; (3) diacylglycerol lipase α/β (DAGL α/β), which catalyzes the biosynthesis of 2-AG; and (4) monoacylglycerol lipase (MAGL), which catalyzes the hydrolysis of 2-AG [17,18,19]. There are other routes of synthesis and degradation involved in the ECS; for a more detailed review, see Cristino et al. (2020) [18].

### 2.1. ECS in the Central Nervous System

In 1988, CB1 was the first cannabinoid receptor described in the CNS [20]. Not long afterward, a second receptor (CB2) was described to be present in peripheral tissue [21] and later in the CNS [13,22,23]. Both are G protein-coupled receptors that link to Gi/o proteins and share 44% of their overall amino acid identity [24]. Their activation inhibits the activity of adenylate cyclase and stimulates the protein kinase activated by mitogenesis (MAPK) as well as phosphoinositide 3-kinase (PI3K). Both receptors are distributed in the presynapses of both excitatory and inhibitory neurons, inhibiting voltage-regulated Ca^2+^ channels and inhibiting the vesicular release of neurotransmitters such as γ-aminobutyric acid (GABA) and glutamate [17,18,25]. This cannabinoid-mediated response is a retrograde inhibitory signaling system that plays an important role in maintaining homeostasis in the CNS [26,27]. Some evidence suggests that CB1s are not only located in the presynaptic membrane. Moreover, studies have shown the presence of CB1s in the membranes of neuronal mitochondria, where they can directly control cellular respiration and energy production [28]. Specifically, activation of mitochondrial CB1s has been shown to inhibit the respiratory chain, decreasing cellular respiration; in this way, they can regulate memory processes [29]. CB1s are also present in astroglial cells in the hippocampus (HPC). Astroglial CB1s have been implicated in the regulation of synaptic transmission, contributing to long-term recognition memory [30]. As mentioned before, CB2 was initially considered to be a peripheral receptor. However, numerous studies have described that CB2s are also present in the brain, mainly in microglia, and that they have an important role in immune modulation. Other studies have shown that CB2s are also expressed in healthy neurons [18,23,31].

Studies have shown that there are marked differences in the presence of both receptors throughout the CNS; specifically, CB1 has shown more presence in the CNS than CB2, but both receptors have been reported to be present in structures such as the hippocampus and prefrontal cortex [32,33]. Additionally, it is important to note that, even though both receptors couple to Gi/o proteins, CB2 has shown more affinity to Gi than to Go proteins [31]. Furthermore, even though CB2 also inhibits voltage-regulated Ca^2+^ channels, it does so with less efficacy than CB1 [34].

Cannabinoid receptors can be activated either by endogenous ligands (endocannabinoids), synthetic cannabinoids, or phytocannabinoids. Regarding the modulation of cannabinoid receptors via phytocannabinoids, it is important to note that CBD has many interesting pharmacological effects. This phytocannabinoid enhances AEA levels and neurotransmission by modulating AEA uptake and metabolism, and it also behaves as an agonist of nonselective cation channels, transient potential vanilloid receptor types 1 and 2 (TRPV1 and TRPV2, respectively) [35,36]. CBD also acts as a nonspecific antagonist of CB1s and CB2s and as an antagonist of G protein-coupled receptor 55 (GPR55), which is a novel cannabinoid receptor [16].

With regard to endocannabinoids, AEA was identified in pig brain samples [37], and 2-AG was extracted from canine intestinal samples [38] and later described in the CNS [39]. These molecules, unlike neurotransmitters, are synthetized and released on demand by membrane phospholipid precursors [40]; they are inhibitory retrograde modulators [18,41], acting as fast retrograde messengers to activate CB1s or CB2s in the presynapse to inhibit neurotransmitter release [41]. Endocannabinoids are rapidly cleared via a process of cellular uptake; then, they are metabolized [42]. The extracellular and intracellular transport mechanisms of endocannabinoids are yet to be fully understood. Some possible routes for AEA and 2-AG transmembrane transport are passive diffusion, endocytosis, movement by transporter proteins, or a combination of these mechanisms [43]. Some evidence suggests that there are cytosolic AEA-binding proteins and that the intracellular distribution of AEA after its uptake is due to specific protein-associated AEA binding activity attributed to heat shock protein 70 and albumin; moreover, fatty acid binding proteins (FABP) 5 and FABP7 have been identified as additional endocannabinoid carriers [43,44,45]. Further evidence has also shown that extracellular AEA and 2-AG transport in the synaptic cleft occurs via microvesicles and not via protein transporters [45].

### 2.2. ECS during Development

There is evidence that shows that there are developmental changes in the ECS [46]. Regarding those changes, it is essential to note that in rodents, brain development undergoes explosive growth from postnatal day (PND) 0 to PND 10 [47]. A study by Hill et al. (2019) analyzed the developmental trajectory of AEA and 2-AG levels in a rodent model at five different early time points in life [48]. They found that in the prefrontal cortex (PFC) and amygdala, AEA levels were almost undetectable at PND 2, slightly increased at PND 12–14 and then increased dramatically by PND 40 and 70. Meanwhile, 2-AG levels at PND 2 are comparable to adult levels, but they dramatically increase on PND 12–14 and decline again to adult levels at PND 40 and PND 70. Finally, in the HPC, both AEA and 2-AG levels were higher at PND 40 and PND 70 than at younger ages, meaning that changes in levels increase in a linear manner in this region [48]. In adolescence, there is enhanced endocannabinoid signaling during the development of the young brain [49]. In particular, animal models have shown that levels of AEA, as well as of the cannabinoid receptor CB1, peak during adolescence and drop in adulthood [40,46,49,50,51]. Additionally, CB1 mRNA has been found to decrease in an age-dependent manner in rats, exhibiting the highest expression during the juvenile (PND 30) period and declining normally throughout adulthood in regions such as the medial PFC, secondary motor cortex, dorsomedial and dorsolateral striatum, dorsal hippocampal regions and ventral subiculum of the HPC [51,52].

The ECS plays a specific role in neural development by controlling the establishment of cortical-subcortical connections [46]. In this regard, a study by Bernabeu et al. (2020) analyzed the postnatal maturation trajectories of layer 5 pyramidal PFC synapses at different postnatal stages [53]. They found that endocannabinoid-mediated long-term depression (LTD) was sexually dimorphic, even though CB1s were functional in both sexes during all developmental stages. This means that this endocannabinoid-mediated LTD first emerged in females during their juvenile period, while in males, endocannabinoid-mediated LTD did not emerge until pubescence. Furthermore, a study by Borsoi et al. (2019) supports this type of dimorphic effect [54]. Single in vivo exposure to WIN 55,212-2 (2 mg/kg), a full CB1/CB2 agonist, ablated the previously mentioned LTD in pubescent (PND 34–37) and adult (PND 90–120) female rats, while in male rats, WIN 55,212-2 did not have this effect.

These findings suggest that the adolescent and prepubertal stages are critical in the development of the ECS, including the production of endocannabinoids and the expression of their CB1/CB2 receptors. In addition, the involvement of the ECS in the maturation of cortical-subcortical connections is clear. Thus, considerable attention should be given to prevent any possible alteration of cannabinoid signaling at the early stages of adulthood.

## 3. Cannabinoid Effect on Neurogenesis

Neurogenesis comprises a series of sequential events that are necessary for the generation of new neurons. The neurogenic process is fundamental for the development of the CNS during the embryonic and early postnatal stages [55,56]. In the past, it was thought that neurogenesis was a specific process that occurred during those early stages of life. However, through the use of different combinations of techniques, including carbon labeling, immunohistochemistry, 5-bromo-2′-deoxyuridine (BrdU) labeling of dividing cells, and other approaches, it was demonstrated that neurogenesis occurs in the adult mammalian brain [57,58,59,60,61,62]. The tightly regulated process of adult neurogenesis [63,64] occurs through the division of neural stem cells (NSCs) (proliferation), their subsequent maturation into neural progenitor cells (NPCs), and their migration (differentiation) to finally mature into neurons (survival) [65,66]. Different and specific molecular markers are present in the cells and characterize those different neurogenesis stages [64,66], as shown in Figure 2. Adult neurogenesis has been found to occur in various mammals (rodents, nonhuman primates, humans) [67] and in certain areas of the brain designated as neurogenic niches: the subgranular zone (SGZ) of the dentate gyrus in the HPC and the subventricular zone (SVZ) in the lateral ventricles [60,62,65]. However, other regions, such as the hypothalamus (HPT), cortex, striatum, habenula, and amygdala, are also considered neurogenic areas in the adult brain [65,68,69]. Adult neurogenesis is one of the plasticity mechanisms in the brain that has been strongly associated with memory formation [70,71,72]. Defects in the neurogenic process have been related to some human neurological and psychiatric diseases [73,74], as well as to cognitive alterations in animal models [75,76,77].

There is significant evidence showing that the modulation of the ECS in the developmental and postnatal stages may lead to alterations in the neurogenic process [78,79]. Herein, we describe the most recent evidence demonstrating that the ECS has a key role in the formation of new neurons, particularly in the adult brain. Although chronic cannabis smoke exposure in mice increased microtubule-binding protein doublecortin (DCX)-positive cell migration in the SGZ, it also promoted a reduction in the number of cells labeled with DCX compared with that in the control group not exposed to cannabis. In addition, chronic cannabis smoke exposure led to an altered morphology of this marker compared with that in the control group [80]. These data are relevant since DCX is a key marker of immature neurons, and its normal expression suggests neurogenesis (Figure 2). In a more refined approach, chronic oral treatment with VCE-003.2 (a cannabigerol-derived cannabinoid acting through PPARγ) improved NSC mobilization and subventricular neurogenesis in mice analyzed by double-labeled BrdU+ and NeuN+ cells in response to mutant huntingtin-induced striatal neurodegeneration [81]. In this respect, 5-bromo-2′-deoxyuridine (BrdU) is a thymidine analog used as a standard technique for cellular proliferation visualization [82], and NeuN is a neuronal nuclei protein widely accepted as a specific cell marker for mature neurons (Figure 2). Similarly, the use of the phytocannabinoid CBD has been useful for studying the effect of the ECS on the production of new neurons. In mice, chronic treatment with this drug led to neurogenic effects in adult rodents, as detected by different markers, such as an increased number of DCX+, BrdU+, NeuN+, and Ki67+ cells [83,84,85]. The latter nuclear protein is associated with cellular proliferation (Figure 2). Importantly, these aforementioned effects are dependent on the amount of CBD administered; thus, a small dose (3 mg/kg) favors the appearance of neurogenic effects, but this does not happen with higher doses (30 mg/kg) [83]. Even when facing challenges such as the model of chronic unpredictable stress (CUS) [84,86], cocaine consumption [85] or bilateral common carotid artery occlusion (BCCAO) [87], CBD restored or promoted some markers of neuronal differentiation (DCX+ cells) and hippocampal neurogenesis (BrdU+NeuN+ cells) in adult mice. As mentioned above, CBD has different mechanisms of action. However, its neurogenic effects are blocked when using specific CB1 and CB2 antagonists/inverse agonists (AM251 and AM630, respectively), thus suggesting that these particular receptors are involved in its effects [84,86]. Table 3 presents detailed information on the effects of CBD on adult neurogenesis.

Regarding CB1, the activation of this receptor by the use of the synthetic agonist arachidonyl-2′-chloroethylamide (ACEA) reversed the impaired adult NPC proliferation induced by the model of forced consumption of ethanol or sucrose in rats with different intensities in the SVZ, SGZ and HPT [88]. Other studies have also confirmed the participation of CB1s in neurogenesis. As an example, mice intranasally (i.n.) administered WIN55,212-2, a full CB1/CB2 agonist, increased the prevalence of BrdU+ cells in the mouse olfactory epithelium, an effect that was blocked by the specific CB1 antagonist/inverse agonist AM251 and that was absent in CB1/CB2 knockout (KO) mice [89]. In an elegant approach, CB1 expression was deleted in NSCs in the adult mouse HPC (nes-CB1 KO), leading to a reduction in the number of those specific cells, as well as DCX+ and BrdU+ cells in the SGZ. The lack of CB1 expression also induced some morphological alterations, such as a reduction in the dendritic length and number of dendritic protrusions. In addition, nes-CB1 KO mice showed alterations in long-term potentiation (LTP) [90], an important form of synaptic plasticity. However, in control animals and rats exposed to acute or repeated administration of cocaine, CB1-specific antagonism/inverse agonism by rimonabant led to an increased number of BrdU+ cells in the SGZ. Although the results of this study are not in accordance with those of previous studies, it is important to mention that the effect reported by Blanco-Calvo et al. (2014) was associated with the prevention of cocaine-induced conditioned locomotion by rimonabant and not a direct effect on neurogenesis [91]. Please refer to Table 4 for details of the experiments that demonstrate the effects of CB1 modulation on neurogenesis.

There is also substantial information highlighting the participation of CB2 in similar neurogenic outcomes (Table 5). When juvenile rats were chronically administered WIN 55,212-2, a full agonist of CB1/CB2 but with high affinity for CB2, the survival of new cells increased in the PFC and striatum, two key terminal fields of the dopaminergic pathway [92]. Moreover, acute i.n. administration of WIN 55,212-2 in mice was enough to increase BrdU+ cells in the olfactory epithelium, an active region that generates neurons through adulthood [89]. However, in other work with rats, juvenile exposure to this drug reduced DCX labeling in the SGZ [93]. Selective CB2 agonists have also led to positive effects on neurogenesis. HU-308 (a CBD derivative that acts as a specific CB2 agonist) treatment in mice induced cell proliferation in the adult HPC, as demonstrated by an increase in BrdU incorporation. This effect was mTORC1-dependent and was accompanied by an augmentation of ribosomal protein S6 phosphorylation (pS6) [94]; both mechanisms have been described as essential elements in neuronal responses to synaptic activity and plasticity [95]. Meanwhile, chronic administration of the selective CB2 agonist JWH-133 in old female mice significantly increased Ki67+ cell and neuroblast migration to the olfactory bulb [96], suggesting proliferation induction by CB2. Under pathological conditions, the activation of CB2 by JWH-133 in rats counteracted the deleterious effect of forced ethanol/sucrose consumption on adult NPC proliferation in the two neurogenic niches SVZ and SGZ [88]. The correct expression of CB2 has been shown to be crucial to the neurogenic effect promoted by its agonist, as the lack of expression of these receptors in CB2 KO mice completely inhibits such effects [89,94].

The use of CB2 antagonists/inverse agonists has also revealed interesting information regarding its involvement in the generation of new neurons in the adult brain (Table 5). Using a model of acute or repeated administration of cocaine in rats, the CB2 antagonist/inverse agonist AM630 induced a restorative effect on hippocampal cell proliferation, which was associated with the prevention of cocaine-induced hyperlocomotion [91]. However, Goncalves and colleagues (2008) showed that JTE907 or AM630, both of which are selective cannabinoid CB2 antagonists/inverse antagonists, reduced cell proliferation in the SVZ of mice [96]. In addition, there was a decline in neuroblast migration to the olfactory bulb prompted by AM630. In a mouse model of CUS, AM630 was shown to be effective in attenuating the proneurogenic effects of CBD in the SGZ. This CB2 antagonism promoted alterations in the number and migration of DCX+ cells, decreased BrdU incorporation, and altered dendritic spine numbers [86]. Similarly, the antagonist JTE907 blocked the JWH-133-induced increase in Ki67+ cells and neuroblast migration in mice [96].

As previously mentioned in this manuscript, cannabinoids can target receptors other than CB1 and CB2 (e.g., GRP55 and TRPV1). The activation of GPR55 by continuous administration of its agonist O-1602 into the HPC promoted early adult hippocampal neurogenesis, as detected by an increase in DCX, Ki67 labeling and BrdU incorporation; the latter effect was blocked in GPR55 KO mice [97]. In contrast, the lack of expression of TRPV1 in TRPV1 KO mice did not disrupt the basal levels of cell proliferation in the SVZ [96], suggesting a distinct involvement in neurogenesis of the different receptors targeted by cannabinoids.

Endocannabinoid synthesis and degradation are crucial for modulating the ECS (Figure 1). Changes in AEA and 2-AG levels produced by the inhibition of their biosynthetic (DAGL for 2-AG) and degrading (FAAH and MAGL, respectively) enzymes have led to exciting results demonstrating its involvement in the generation of new neurons in adulthood (Table 6). The ~80% reduction in brain 2-AG levels in DAGL-α KO mice was directly associated with the significant reduction in BrdU incorporation in SVZ cells [98]. Similarly, the intracerebroventricular (i.c.v.) administration of tetrahydrolipstatin (THL) and RHC-80267, both selective DAGLα/β inhibitors, reduced the proliferation of progenitor cells in young mice, as observed by changes in the number of neuroblasts migrating from the SVZ to the olfactory bulb. In addition, RHC-80267 decreased the number of Ki67+ cells [96]. However, Rivera and colleagues (2015) showed that the pharmacological inhibition of FAAH by URB597, which limits AEA degradation, can promote an increase in the number of subventricular and hippocampal proliferative cells in rats when they are administered a single dose, but the opposite effect is observed when URB597 is administered as a repeated treatment [99]. However, other studies have shown similar proneurogenic effects in the SVZ of mice, even when URB597 was administered repetitively [96]. Under pathological conditions, stimulation of the ECS has led to compelling results. Chronic constriction injury to the sciatic nerve in adult rats reduces hippocampal brain-derived neurotrophic factor (BDNF) mRNA, and decreases the number of proliferating cells and survival of newly mature neurons in the HPC. However, chronic treatment with the systemic FAAH inhibitor URB597 restored these cellular deficits in rodents with this type of lesion [100]. Conversely, in a model of forced consumption of ethanol/sucrose liquid diets in rats, treatment with the same drug to modulate FAAH activity did not show beneficial effects on adult NPC proliferation in either the SVZ or SGZ [88]. Interestingly, a treatment aimed to raise endocannabinoid concentrations by acute i.n. administration of the combination of URB597 (an FAAH inhibitor) and JZL184 (an MAGL inhibitor) in mice successfully increased BrdU+ cells in the olfactory epithelium [89]. These data correlate with the restorative effect of JZL184 in the CUS model of depression in mice. In this case, the chronic administration of this MAGL inhibitor restored not only the number of BrdU+ and DCX+ cells in the dentate gyrus but also hippocampal LTP, an important indicator of brain plasticity [101].

The overall information discussed in this section suggests the involvement of the different members of the ECS in maintaining and promoting neurogenesis in the adult brain. The fact that CBD administration, in most of the cases (Table 3), stimulates adult neurogenesis markers suggests a feasible use of this substance to promote the generation of new neurons during adulthood. However, great caution and more studies are necessary, as the dose of CBD seems to be critical in stimulating those effects [83]. Additionally, although the antagonism of CB1 and CB2 reversed the CDB neurogenic effects, suggesting the participation of those receptors, the other CBD mechanisms of action (see Section 2 and Figure 1) must be further studied to identify their possible participation and even interference in the observed neurogenic outcomes.

Regarding the direct participation of CB1 and CB2 in promoting neurogenesis, in many cases under pathological conditions (e.g., cocaine, sucrose/ethanol consumption, CUS), treatment duration is an important factor in outcomes, as demonstrated in Table 4 and Table 5. Most of the studies that reveal a neurogenic effect mediated by the stimulation of cannabinoid receptors involve chronic treatment with agonists. This makes sense, as the neurogenic process has been described to be a tightly regulated process [65,66]; for that reason, a strong stimulus may be necessary to disrupt this regulation. Additionally, it should be noted that the ECS is strongly associated with the modulation of GABAergic and glutamatergic neurotransmission [17,18,25]. Therefore, these synaptic mechanisms may participate in the neurogenic effect reported for CB1/CB2 stimulation.

The data summarized in Table 6 also corroborate the implications of cannabinoid signaling in neurogenesis. Based on all the information collected, we can assure that the neurogenic effect can vary between rodent species, their method of administration, and the different neurogenic niches. However, we can certainly ensure that strict control and equilibrium of the ECS are necessary to maintain optimal adult neurogenesis. Additionally, under pathological conditions (e.g., Alzheimer’s disease, Parkinson’s disease, Huntington disease, and major depressive disorder) that affect the generation of new neurons in adulthood [102,103], the stimulation of cannabinoid signaling seems to be a feasible option for rescuing and restoring the process of neurogenesis. The clinical scope of this information is still missing from basic science, but the expectation is undoubtedly largely because of ample evidence. Hopefully, we will have this information in the near future.

## 4. Effect of Cannabinoids on Behavioral Processes: Stress, Anxiety, Learning and Memory

Adult neurogenesis occurs in specific regions of the brain and has been linked to the modulation of different behaviors and vice versa [104]. Importantly, as mentioned in the previous section, there is plenty of evidence of the regulatory role of ECS signaling in adult neurogenesis. Thus, we considered that a section describing the modulatory role of ECS on some behaviors would be relevant.

The ECS is a crucial modulatory system allowing an organism to adapt to its changing environment. In recent years, a large body of data has emerged demonstrating the crucial role of the ECS in regulating diverse brain functions and behaviors, such as alcohol/cocaine consumption [105], sexual behavior [106], and feeding [107]. Importantly, Goldstein Ferber et al. (2019) suggested that adolescence is a critical period of brain maturation and development that is vulnerable to perturbations induced by cannabis exposure [40]. In the following paragraphs, we describe the effect of ECS modulation on anxiety (Section 4.1), stress (Section 4.2), learning and memory (Section 4.3).

### 4.1. Anxiety

Anxiety is a normal and adaptive response of animals that promotes harm avoidance. However, excess anxiety can trigger serious psychological and behavioral problems, such as negative affect, autonomic symptoms, increased vigilance and passive avoidance [108]. It is well known that corticolimbic structures are critical for the regulation of fear and anxiety. We previously mentioned the presence and participation of cannabinoid receptors in the development of some cortical regions; however, in adolescence and adulthood stages, the ECS faces other dynamic states. CB1s are densely expressed in corticolimbic brain regions such as the PFC in adolescent rats [109]. For example, CB1 expression in the PFC decreased in male adolescent rats [110]. However, dynamic changes in ECS signaling during adolescence parallel normative changes in corticolimbic circuitry [111]. Heng et al. (2011) showed that CB1-mediated signaling decreases during development and regulates limbic/associative cortical areas in rats [112].

It has been suggested that the ECS regulates anxiolytic- and anxiogenic-like effects in adults. In this respect, anxiolytic-like effects were observed in wild-type mice after the administration of a low dose (1 μg/kg) of the CB1 agonist CP55,940. Meanwhile, no effect was observed in knockout mice lacking CB1 in cortical glutamatergic neurons. In the same manner, knockout mice lacking CB1 receptors on the GABAergic terminals in the forebrain mediated an anxiogenic-like effect under a high dose (50 μg/kg) of the CB1 agonist CP55,940 [113]. Additionally, when CB1 expression in glutamatergic neurons is restored in knockout mice, after a genetic strategy to partially reconstitute wild-type CB1 receptor functions, some of the anxiety symptoms disappear [114]. Some studies involving repeated agonism during adolescence have provided support for CB1 contributions in this matter. In particular, repeated CBD administration produced anxiolytic effects detected in mice as increased time spent in the open arms in the elevated plus maze test [85]. However, Keeley et al. (2015) showed that chronic administration of THC in adolescent rats modifies HPC structure and indicates deficits in memory and anxiety task performance [115,116]. Notably, in a study by Renard et al. (2017), it was shown that THC exposure in adolescent rats significantly increased anxiety levels (decreased exploration time of social motivation and social recognition), suggesting that adolescence represents a selective neurodevelopmental window of vulnerability in the developing brain that is particularly sensitive to the effects of chronic THC exposure [117]. Additionally, chronic exposure to THC in adolescent mice increased the percentage of shredding in the nestlet shredding task [118] and immobility times in adult rats during the forced swim test [119]. More recently, a study by De Gregorio et al. (2020) showed that chronic exposure to low-dose THC in adolescent rats leads to persistent behavioral abnormalities related to some but not all aspects of depressive reactivity [120]. In particular, THC administration in rats induced some anxiety behaviors (fewer entries into the open arms of the elevated plus maze) and altered serotoninergic neuron firing rate activity [120]. In another study, it was shown that rats treated with chronic THC during adolescence showed a higher level of anxiety-like behaviors, which led to a significant reduction in food intake and body weight [121]. In this regard, Silva et al. (2016) studied the effect of early life experience on THC exposure in adolescent rats, suggesting that chronic THC treatment during adolescence produces an anxiolytic-like effect, and the prepubertal period may represent a particular period of sensitivity to THC [122].

Regarding the modulation of the ECS by synthetic drugs, Renard et al. (2016) showed that cannabinoid receptor activation in rats by the synthetic cannabinoid CP55,940 during adolescence induced long-lasting changes in the PFC structure and function in adulthood that may underlie cognitive deficits in adulthood, such as low social motivation and social cognition [123]. Additionally, the researchers showed that CB1 overactivation with CPP55,940 during adolescence interfered with normal CB1-mediated developmental processes, thereby leading to persistent alterations in the homeostasis of the GABA/glutamate balance in the PFC. In addition, chronic exposure to high doses of WIN55,212-2, a full CB1/CB2 agonist, in adolescent rats induced anxiety-like effects (increasing latency to feed) in adulthood, as measured by the novelty-suppressed feeding test [124]. Additionally, later studies reported that repeated CB1 antagonism/inverse agonism (AM251) in adolescence increased social interactions, increased the expression in the PFC of the glutamic acid decarboxylase 67 (GAD67, an enzyme that catalyzes the synthesis of GABA), and reduced hippocampal CB1 expression in female rats, with no effects observed in males [109]. Additionally, acute administration of the FAAH pharmacological inhibitor URB597 or the MAGL inhibitor KML29 decreased anxiety-like behaviors in adult rats, whereas AM251, a CB1 antagonist/inverse agonist, blocked these effects [125].

This section demonstrates that the administration of CB1 agonists or antagonists in adolescent rats can modify the mature brain, particularly limbic/associative cortical areas, leading to modifications in behaviors such as anxiety. Additionally, the long-term effects of perturbation to the ECS will vary depending on the timing and duration of exposure [111]. Although several studies show that administration of THC during adolescence modifies anxiety behavior, its effect is inconsistent; in this regard, increases [117,118] and decreases [119,120,121] in anxiety levels have been reported. These inconsistent results can be explained by the differences in the doses, time of administration, test behaviors assessed after the final administration, and model animal. Therefore, it is necessary to elucidate the effect of the administration of CB1 agonists in adolescents and the impact on anxiety, such as its effects leading to modifications of brain function in the long term and even in adulthood.

It is important to mention that, as reviewed in the previous sections, there is evidence that shows that developmental changes occur in the ECS [46]. Specifically, animal models have shown that AEA, 2-AG and CB1s reach maximum levels during adolescence; then, in adulthood, their levels decrease [40,46,48,49,50,51]. Thus, the results presented in this section are consistent with the hypothesis that adolescence is a sensitive period during which the developmental trajectory is malleable; furthermore, these results contribute to an understanding of the role of the ECS in adolescent development [126].

### 4.2. Stress

Stress is any intrinsic or extrinsic stimulus that evokes a biological response, and any compensatory response to these stimuli is known as a stress response [127]. The ECS modulates the neuroendocrine and behavioral effects of stress [128] and is also capable of being affected by stress exposure itself [40]. Additionally, the stress and reward networks are highly interactive, and the ECS may modulate such interactions.

In this regard, single and repeated CBD administration in mice produces antidepressant-like effects evident in the reduced immobility time observed in animals in the tail suspension test [83]. Additionally, Fogaça et al. (2018) suggested that chronic CBD administration in mice produced anti-stress effects, as this drug decreased anxiety and the negative outcomes in the novelty-suppressed feed test [86]. This work associates those effects with the reduced expression of FAAH in a cannabinoid receptor-dependent manner. Additionally, it has been shown that cannabis exposure in mice significantly increases self-grooming behavior in animals during the open field test, even when animals are exposed to a model of restraint stress [80]. Another study aimed to determine whether increasing doses of a CB1 agonist (HU-210, 25, 50 and 100 μg/kg) administered in adolescent rats would affect stress in adulthood [129]. They showed that these doses increased stress responsivity (increasing peak corticosterone levels) in adult males more than in female rats.

Additionally, several studies have investigated the effects of repeated administration of CB1 agonists or antagonists/inverse agonists during adolescence. Alteba et al. (2016) demonstrated that the stimulation of CB1/CB2 by WIN55,212-2 during late adolescence can reverse the long-term effects of early stress on emotional behavior and short-term memory in both male and female rats [130]. Regarding the antagonism of cannabinoid receptors, Lee et al. (2015) showed that during peri-adolescence in male rats, CB1 blockade by AM251 increased active stress-coping behavior in the forced swim test and moderately increased risk assessment behavior in the elevated plus maze [131]. Moreover, Simone et al. (2018) reported a lack of interaction between treatments (repeated stress and AM251 during adolescence) in measures of emotional behaviors and stress in adult rats [109]. Specifically, female rats treated with AM251 had reduced CB1 expression and increased GAD67 expression in the HPC and increased social interactions. Furthermore, Simone et al. (2018) showed that the inhibition of CB1 by AM251 promoted social interaction in females, while there were no effects in males. This is evidence of sex-specific vulnerability to the effects of alterations in the ECS [126]. In summary, the administration of a CB1 agonist shows that increasing stress reactivity [80] also suppresses adult neurogenesis [129]; in the same way, the administration of an antagonist of CB1 increases active stress coping [131] and social interaction [109]. These results show that homeostatic regulation of the ECS is necessary; as it increases or decreases, this system can modify neuroendocrine and behavioral stress.

The information presented in this section supports the proposal of Surkin et al. (2018), namely, the ECS could be part of a negative feedback system that limits the acute neuroendocrine stress response [132]. Additionally, the ECS has been associated with neuroendocrine modulation in several stages of life, particularly in adolescence, when the ECS is highly sensitive to pharmacological manipulation.

### 4.3. Learning and Memory

Learning is understood as changes in the behavior of an organism that result from regularities in the environment of the organism [133], while memory consists of the capacity to encode, store, consolidate, and retrieve information [134]. It is well known that the HPC, PCF and amygdala, among other structures, are involved in numerous processes, such as memory and plasticity [135]. It is also well known that these regions exhibit high CB1 expression [136]. Several studies have highlighted that adolescent cannabinoid exposure persistently impairs memory.

In this regard, Lujan et al. (2018) showed that repeated CBD administration increases the discrimination index of mice in a novel object recognition task and attenuates cocaine-induced conditioned place preference [85]. Additionally, repeated exposure to THC during adolescence has been associated with impairments in mice during a novel object recognition memory task [118]. Additionally, Chen and Mackie (2020) recently showed that mice treated with THC during adolescence acquired proficiency in a working memory task more slowly than vehicle-treated mice [137]. Another study reported that adolescent THC exposure in rats induced deficits in recognition memory, reduced GAD67 levels, and reduced basal GABA levels within the adult PFC, suggesting that THC exposure during adolescence disturbs the physiological maturation of the GABA system in this brain region [138]. Furthermore, Gibula-Tarlowska et al. (2020) showed that adolescent rats that were administered THC in combination with ethanol showed more potent deficits in spatial learning and memory and cognitive flexibility (reversal learning) [139]. These studies show that the behavioral picture triggered by adolescent THC administration is more complex than previously known. Renard et al. (2016) postulated that overactivation of CB1s by exogenous cannabinoids (such as THC) during adolescence could interfere with normal CB1-mediated developmental processes, thereby leading to persistent alterations in the homeostasis of the GABA/glutamate balance in the PFC [140]. This could lead to impaired synaptic and structural plasticity in brain regions that play crucial roles in learning and memory. More recently, it was reported that chronic administration of increasing amounts of THC (1.0, 1.5, and 2.0 mg/15 mL THC gelatin over 33 days) in adolescent male rats produced impaired Pavlovian reward-predictive cue behaviors; these behaviors occurred in parallel with the loss of CB1 in the glutamatergic terminals of the ventral tegmental area in males during adulthood [141]. This study demonstrated that voluntary oral consumption of THC during adolescence leads to alterations in reinforcement learning processes. Poulia et al. (2019) showed that low-dose THC (0.3 mg/kg) exposure led to increased spontaneous locomotor activity, impaired behavioral motor habituation and defective short-term spatial memory in adolescent rats; these outcomes paralleled decreased BDNF protein levels in the PCF [142]. Furthermore, Stringfield and Torregrossa (2021) showed that adult rats that self-administered THC in adolescence showed reductions in multiple proteins involved in synaptic transmission, as well as reductions in cannabinoid receptors in regions of the brain, such as the PFC, that undergo developmental changes during adolescence [143]. However, contrary to other studies, in this study, adolescent self-administration of THC did not produce memory deficits.

However, chronic treatment with the synthetic cannabinoid CP55,940 in rats not only induced dysfunction of PFC network activity but also disrupted interactions between the HPC and PFC [144]. In this regard, Renard et al. (2016) demonstrated in rats that chronic exposure during adolescence to the same drug (CP55,940) led to long-lasting structural and functional changes in adulthood [123]. These changes in the medial PFC include impaired HPC–PFC synaptic plasticity and significantly decreased expression of postsynaptic density protein 95 (PSD95, a regulator of synaptic maturation that is used as a postsynaptic marker).

Additionally, Cass et al. (2014) indicated that early (PND 35–40) and mid-adolescence (PND 40–45) in rats constitutes a critical period during which repeated CB1 stimulation is sufficient to elicit an enduring state of PFC network disinhibition resulting from a developmental impairment of local prefrontal GABAergic transmission [145]. This may explain persistent deficits of local prefrontal GABAergic transmission and cortical synaptic plasticity. In this regard, Abboussi et al. (2014) showed that the chronic stimulation of cannabinoid receptors by WIN55,212-2 (CB1/CB2 agonist) in adolescent rats induced spatial learning and memory deficits [93]. Moreover, it was demonstrated that adolescent treatment with WIN55,212-2 increased acoustic startle latency and novel object exploration in rats [92].

In summary, there are multiple behavioral effects of cannabinoid exposure during adolescence, suggesting the involvement of the ECS in behavior and signifying that the magnitude of the differences depends on the age of consumption onset, dose, and length of exposure [146]. Table 7 summarizes the effects of ECS modulation on behavior. These reports suggest that modulation of the ECS in adolescents alters functional and structural plasticity and impairs learning and memory processes.

## 5. Conclusions

As demonstrated in the present review, the ECS is a crucial modulatory system for the development of the nervous system, with the adolescent–young adulthood stage being a particularly sensitive time period. Exposure to cannabinoids during this period can have long-term consequences, such as adult neurogenesis and behavior, through to adulthood. Furthermore, exposure to cannabinoids interferes with the optimal performance of the ECS and provokes defects in the neurogenic process; these outcomes have been associated with psychiatric diseases and cognitive alterations. Thus, ECS equilibrium is necessary to maintain optimal adult neurogenesis. The present review has provided multiple lines of evidence showing that there are several behavioral effects of cannabinoid exposure during adolescence, demonstrating the effect that adolescent exposure to cannabinoids has in adulthood, and indicating that the magnitude of the effects depends on important factors such as age of consumption onset, dose and length of exposure, as summarized in the tables.

Although marijuana (cannabis) is the most consumed illegal drug in the world, it is becoming legalized in an increasing number of countries. Additionally, its highest consumption rate is observed during adolescence, so it is of particular importance to understand its effects in the early stages of life (adolescence and young adulthood), particularly the alterations it can provoke in brain functionality, along with its impact on adult neurogenesis and the optimal performance of adaptive behaviors (e.g., the stress response, learning and memory, and anxiety). The legalization of marijuana implies not only changing its status from illegal to legal, but studies have shown that there is an immediate impact on its consumption, as well as on its perceived availability and risk of use. It is therefore important to continue research on the association between cannabinoid use and all its possible beneficial or detrimental effects, including adult neurogenesis and adaptive behaviors. This research will also help improve the present understanding of the consequences that the legalization and use of marijuana and cannabinoids can have on the world population. Some issues to investigate in-depth in future studies are: the mechanisms by which adolescent cannabinoid consumption induces sex-specific plastic differences; the role of noncannabinoid receptors, such as TRPV1, GPR55 and PPARγ, in the modulation of adult neurogenesis mediated by cannabinoids; and the specific mechanisms and intracellular pathways involved in cannabinoid-induced neuroplastic regulation. Additionally, whether the facilitation of neuronal proliferation, differentiation, migration, maturation, and neuronal survival mediated by the ECS can be effective under pathological conditions warrants further investigation. We know that the scientific community focused on studying adult neurogenesis and cannabinoids is striving to address the above and many other issues that remain unresolved. Joint effort in this area will help improve the present understanding of ECS function, cannabinoids, and their neuroplastic and behavioral implications. This research will undoubtedly yield new treatment opportunities for neurodevelopmental, neuropsychiatric, and addiction disorders.

## Figures and Tables

**Figure 1 ijms-22-07450-f001:**
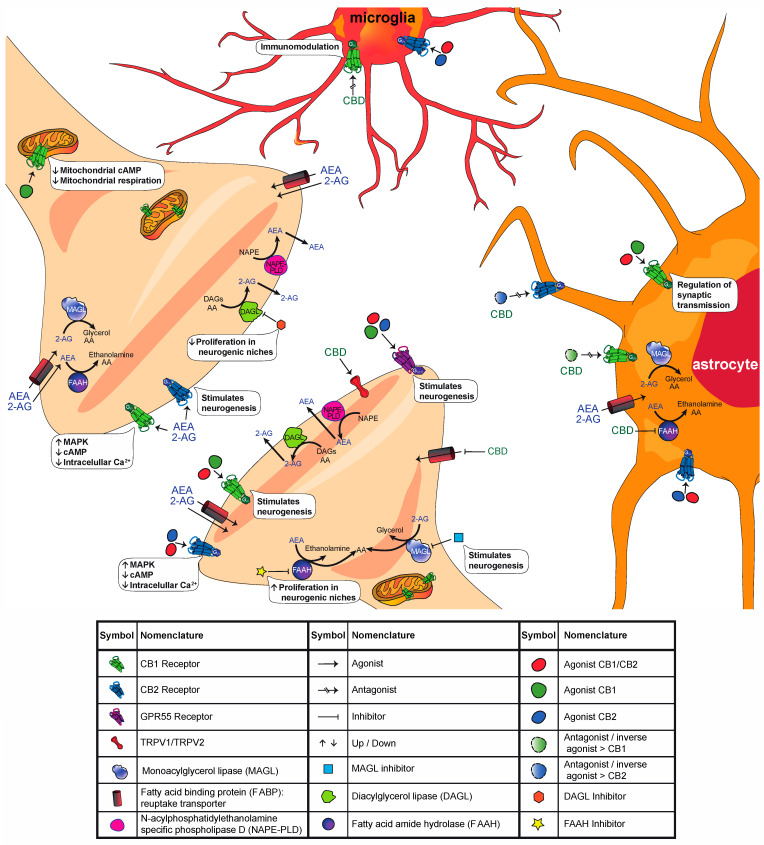
Schematic representation of cannabinoid function at the synapse. We depict the different components of the ECS (receptors, endogenous ligands, transporters across the membrane, biosynthetic and degradative enzymes) in neurons, astrocytes, and microglia. Some key effects elicited by the modulation of the ECS are highlighted. For a list of agonists, antagonist/inverse agonists and inhibitors, see Table 2. For more details, refer to the main text.

**Figure 2 ijms-22-07450-f002:**
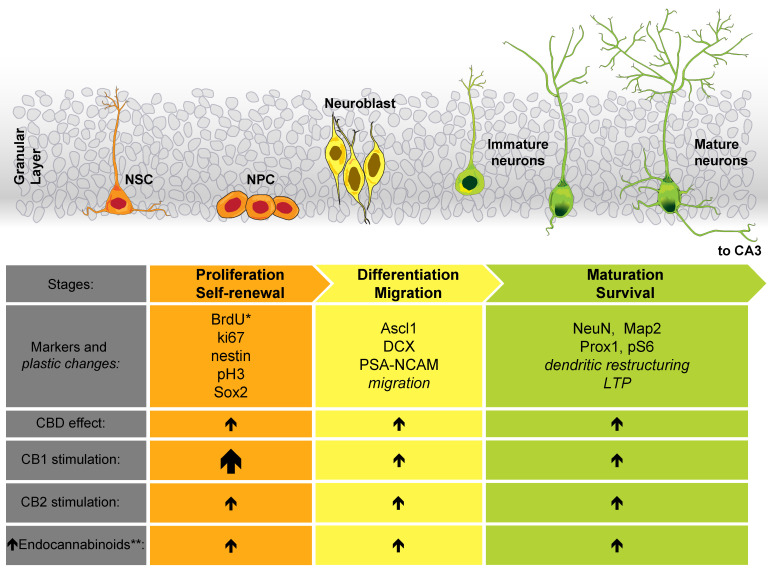
Participation of the ECS in adult hippocampal neurogenesis. We illustrate the distinct cell morphologies associated with the different stages of adult hippocampal neurogenesis at the top of the figure. Below, we include some of the molecular markers and plastic changes (italics) associated with the specific neurogenic stages. This list is not exhaustive; it mainly includes the markers referred to in the text. In the last rows, we show the effect of stimulating the ECS (a larger arrow represents a major number of references supporting this effect). * As mentioned in the text, BrdU is not an endogenous molecular marker of neurogenesis. ** To increase the concentration of endocannabinoids, we primarily refer to the inhibition of their degrading enzymes. Abbreviations (not described in the main text): Achaete-scute family BHLH transcription factor 1 (Ascl1), microtubule associated protein 2 (MAP2), phospho-histone 3 (pH 3), polysialylated-neural cell adhesion molecule (PSA-NCAM), prospero homeobox 1 (Prox1), and SRY-box transcription factor 2 (Sox2).

**Table 1 ijms-22-07450-t001:** Consumption of the main drugs of abuse (% of the 15-year-old and above population).

Substance	Worldwide	Americas	Oceania	Africa	Asia	Europe
Alcohol [1]	43.00	54.10	53.80	32.20	33.10	59.90
Tobacco [2]	23.40	15.00	33.50	17.30	43.70	31.20
Cannabis [3]	3.80	8.80	10.57	6.32	1.86	5.39
Amphetamines [3]	0.55	1.30	1.35	0.41	0.42	0.47
Opioids [3]	1.16	1.86	2.47	1.04	1.11	0.68
MDMA (Ecstasy) [3]	0.41	0.53	1.67	0.26	0.37	0.61
Cocaine [3]	0.38	1.49	1.56	0.27	0.06	0.89

**Table 2 ijms-22-07450-t002:** Endogenous and synthetic modulators of the ECS.

Mechanism of Action	Compounds
Agonist CB1/CB2	2-AG *, AEA *, Δ9-THC **, HU-210, WIN 55,212-2, CP55,940
Agonist CB1	ACEA
Agonist CB2	AM1241, HU-308, JWH-133, JWH-015, JWH-056
Antagonist/inverse agonist > CB1	AM251, ibipinabant, LY320135, AM28, rimonabant (SR141716A), surinabant
Antagonist/inverse agonist > CB2	AM630, JTE907, SR 144528
DAGL Inhibitor	RHC-80267, tetrahydrolipstatin(THL)
FAAH Inhibitor	URB597, URB937
MAGL inhibitor	JZL184, KLM29
CB1/CB2 weak antagonist TRPV1 and TRPV2 agonist GPR55 antagonist FABP inhibitor FAAH inhibitor	CBD **

Note: * endocannabinoid; ** phytocannabinoid.

**Table 3 ijms-22-07450-t003:** CBD effects on neurogenesis.

Animal Model ♂ * (Age in PND)	Dose Duration	Niche Analyzed	Markers	Effects	References
Mouse B6 ** + chronic unpredictable stress (CUS) (PND: 84)	30 mg/kg i.p. 14 days	dorsal HPC	BrdU NeuN	CBD alone: ↑ BrdU+ cells, DCX+ cells, BrdU+NeuN+ cells CBD restored CUS-induced effects (↑ BrdU+, DCX+ cells)	[84]
mouse B6 + CUS (PND: 58)	30 mg/kg i.p. 14 days	dorsal HPC	BrdU DCX Dendritic spines	CBD restored CUS-induced effects (↑ DCX+ cells and their migration, BrdU+ cells, NeuN+BrdU+ cells, number of dendritic spines and their length/branch number)	[86]
mouse CD1 + cocaine (PND: 42)	20 mg/kg i.p. 10 days	dorsal HPC	BrdU DCX NeuN	CBD alone: ↑ BrdU+NeuN+ cells CBD restored cocaine-induced effects (↑ BrdU+NeuN+ cells, DCX+ cells)	[85]
mouse B6 + bilateral common carotid artery occlusion (BCCAO) (PND: 70)	10 mg/kg i.p. 3 days	dorsal HPC	DCX	CBD restored BCCAO-induced effects (↑ DCX+ cells, dendritic restructuring)	[87]
mouse Swiss albino (PND: 42)	3, 30 mg/kg i.p. 15 days	dorsal HPC SVZ	BrdU DCX Ki67	3 mg/kg: ↑ BrdU+ cells, DCX+ cells, Ki67+ cells 30 mg/kg: ↓ BrdU+ cells, DCX+ cells, Ki67+	[83]
mouse B6 + AAV-mediated expression of mutant huntingtin in striatum (PND: 70)	VCE-003.2 *** 10 mg/kg p.o. 18 days	SVZ	Ascl1 GFAP Ki67	VCE restored the huntingtin-induced effects (↑ GFAP+Ki67+ cells, Ascl1+ cells mobilization)	[81]
		striatum	BrdU DCX NeuN	(↑ DCX+ cells, NeuN+BrdU+ cells)	

Note: Increase (↑), decrease (↓), male(♂). * During the bibliographic review, we found studies that exclusively used male mice in their experiments. ** The C57BL/6 mouse strain is referred to as B6 in this and the following tables. *** VCE-003.2 is a cannabinoid acting through PPARγ. It was included in this table because, as with CBD, its mechanism of action is mainly CB1/CB2-independent. GFAP: Glial fibrillary acidic protein is expressed by progenitors cells and mature astrocytes.

**Table 4 ijms-22-07450-t004:** Effects of CB1 modulation on neurogenesis.

Animal Model ♂ * (Age in PND)	Drug	Drug Category	Dose Duration	Niche Analyzed	Markers	Effects	References
rat Wistar + forced consumption of ethanol or sucrose (PND: 77)	ACEA	CB1 agonist	3 mg/kg i.p. 5 days	dorsal HPC HPT SVZ	BrdU pH3	ACEA restored the forced consumption-induced effects HPC: ↑ pH3+ cells HPC, HPT, SVZ: ↑ BrdU+ cells	[88]
rat Wistar + cocaine (PND: 77)	rimonabant	CB1 antagonist/inverse agonist	3 mg/kg i.p. 1 day	dorsal HPC SVZ	BrdU GFAP	Those effects are exerted by ribonabant alone or + cocaine SVZ: ↓ BrdU+ cells HPC: ↑ BrdU+ cells, GFAP+ cells	[91]
mouse B6 + CUS (PND: 58)	AM251	CB1 antagonist/inverse agonist	0.3 mg/kg i.p. 14 days	dorsal HPC	BrdU DCX Dendritic spines	Block the neurogenic effect of CBD (↓ DCX+ cell migration, BrdU+ cells, and spines)	[86]
mouse Swiss Webster or CB1/CB2 KO mouse (PND: 49)	AM251	CB1 antagonist/inverse agonist	50 μL, 10 μM/mouse i.n. 1 day	olfactory epithelium	BrdU	In WT: Blocked the neurogenic effect of WIN55,212-2 (↓ BrdU+ cells) In KO: No changes	[89]
mouse nestin-CB1 KO (neuronal stem cells CB1 KO) (PND: 56)	No pharmacological treatment			dorsal HPC	BrdU DCX Dendritic spines NeuN Nestin LTP	At 28 and 56 dptm (days post-tamoxifen): ↓ nestin+ cells, DCX+ cells, NeuN+ cells, BrdU+ cells At 28 dptm: ↓ dendritic length and dendritic protrusions. Altered LTP	[90]

Note: Increase (↑), decrease (↓), male(♂). * During the bibliographic review, we found studies that used exclusively male mice in their experimentation.

**Table 5 ijms-22-07450-t005:** Effects of CB2 modulation on neurogenesis.

Sex, Animal Model (Age in PND)	Drug	Drug Category	Dose Duration	Niche analyzed	Markers	Effects	References
(sex not specified) mouse B6 or CB2 KO mouse (PND: 56)	HU-308	CB2 agonist	15 mg/kg i.p. 5 days	dorsal HPC	BrdU Nestin pS6	B6: ↑ BrdU+ cell, pS6+ cells, BrdU+pS6+cells, Nestin+pS6+ cells KO: No changes	[94]
♂ rat Wistar + forced consumption of ethanol or sucrose (PND: 77)	JWH133	CB2 agonist	0.2 mg/kg i.p. 5 days	dorsal HPC HPT SVZ	BrdU pH3	Restored the forced consumption-induced effects in HPC, HPT, SVZ: ↑ BrdU+ cells, pH3+ cells	[88]
♂ rat Wistar (28 or 56)	WIN55,212-2	CB2 > CB1 agonist	1 mg/kg 20 days	dorsal HPC ventral HPC	DCX	28 PND: ↓ dorsal DCX+ cells	[93]
♂ rat Lewis (PND: 42)	WIN55,212-2	CB2 > CB1 agonist	2 mg/kg i.p. 14 days	PFC striatum SVZ	BrdU	PFC, striatum, SVZ: ↑ BrdU+ cells	[92]
♀ mouse B6 (PND: 42 or 168)	JWH-133	CB2 agonist	0.6 mg/kg i.p. on days 1–3 0.9 mg/kg i.p. on days 4–7 1.2 mg/kg i.p. on days 8–10	olfactory bulb (OB) SVZ	BrdU Ki67 Neuroblast migration to OB	JWH: ↑ Ki67+ cells, neuroblast migration to OB	[96]
	AM630 JTE907	CB2 antagonist/inverse agonist	5 mg/kg i.p. 5 days			AM: ↓ Ki67+ cells, neuroblast migration to the OB JTE: ↓ Ki67+ cells JTE: Block the neurogenic effect of WIN55,212-2 and JWH (↓ Ki67+ cells, neuroblast migration)	
♂ rat Wistar + cocaine (PND: 77)	AM630	CB2 antagonist/inverse agonist	3 mg/kg i.p. 1 day	SVZ dorsal HPC	BrdU GFAP	Those effects are exerted by AM630 alone or + cocaine SVZ: ↓ BrdU+ cells HPC: ↑ BrdU+ cells, GFAP+ cells	[91]
♂ mouse B6 + CUS (PND: 58)	AM630	CB2 antagonist/inverse agonist	0.3 mg/kg i.p. 14 days	dorsal HPC	BrdU DCX Dendritic spines	Block the neurogenic effect of CBD (↓ DCX+ cells, DCX+ cells migration, NeuN+BrdU+ cells, spines)	[86]

Note: Increase (↑), decrease (↓), male(♂), female (♀).

**Table 6 ijms-22-07450-t006:** Effects of ECS enzyme modulation on neurogenesis.

Sex, Animal Model (Age in PND)	Drug	Drug Category	Dose Duration	Niche Analyzed	Markers	Effects	References
♀ mouse B6 (PND: 42 or 168)	RHC-80267 THL	DAGL inhibitor	RH = 0.01 μg or 0.3 μg i.c.v. THL = 0.15 μg i.c.v q.a.d. 7 days	OB SVZ	BrdU Ki67 Neuroblast migration to OB	RHC: ↓ Ki67+ cells RHC + washout: Partial recovery of Ki67+ cells RHC or THL: ↓ neuroblast migration	[96]
DAGL KO mouse (sex and age not specified)	No pharmacological treatment			dorsal HPC	BrdU	↓ BrdU+ cells	[98]
♂ mouse Swiss Webster or CB1/CB2 KO mouse (PND: 49)	JZL184 + URB597	MAGL inhibitor + FAAH inhibitor	JZ = 50 μL, 10 μM/mouse i.n. 1 d URB = 50 μL, 100 μM/mouse i.n. 1 day	Olfactory epithelium	BrdU	In WT: ↑ BrdU+ cells In KO: No change	[89]
♂ mouse B6 + CUS (PND: 63)	JZL184	MAGL inhibitor	8 mg/kg i.p.q.a.d. 3 weeks	dorsal HPC	BrdU DCX LTP	JZL restored the CUS -induced effects (↑ BrdU+ cells, DCX+ cells and LTP)	[101]
♂ rat Wistar + chronic constriction injury (CCI) (PND: 49)	URB597	FAAH inhibitor (systemic)	5.8 mg/kg i.p. 14 days	dorsal HPC	BrdU BDNFmRNA Ki67	URB597 restored the CCI-induced effects (↑ BrdU+ cells, Ki67, BDNFmRNA)	[100]
	URB937	FAAH inhibitor (peripheral)	1.6 mg/kg i.p. 14 days			No change	
♂ rat Wistar (PND: 77)	URB597	FAAH inhibitor	0.3 mg/kg i.p. 1 day	dorsal HPC HPT SVZ	BrdU pH3	SVZ: ↑ pH3+ cells HPT, SVZ: ↑ BrdU+ cells	[99]
			0.3 mg/kg i.p. 5 days			HPC, HPT: ↓ pH3+ cells HPC: ↓ BrdU+ cells and its survival	
♂ rat Wistar + forced consumption of ethanol or sucrosesucrose (PND: 77)	URB597	FAAH inhibitor	0.3 mg/kg i.p. 5 days	dorsal HPC HPT SVZ	BrdU pH3	No effect	[88]

Note: Increase (↑), decrease (↓), male (♂), female (♀).

**Table 7 ijms-22-07450-t007:** Effects of the ECS modulation on behavior.

Sex, Animal Model (Age in PND)	Drug	Drug Category	Dose Duration	Behavioral Effects	References
♂ rat Sprague-Dawley (30)	THC	CB1/CB2 agonist	1 mg/kg i.p. 20 days	↓ Spent time in open arms (EPM), latency to immobility (FST), sucrose preference (SPT) = Latency to feed (NSFT), distance traveled (OFT)	[120]
♂♀ mouse B6 (28)			3 mg/kg i.p. 21 days	↓ Impaired performance (delayed alternating T-maze) = Social behavior, open arm entries (EPM) and decision making (T-maze)	[137]
♂ rat Wistar (30)			1 mg/kg i.p. 4 days	↑ Primary latency (Barnes maze) = Horizontal locomotor activity test	[139]
♂ mouse CD1 (28)			3 mg/kg, i.p. 20 days	↑ % shredded (nestlet shredding task), marbles buried (marble burying task) ↓ Discrimination index (NORT), open arm entries (EPM) = Total distance traveled (OFT)	[118]
♂♀ rat Sprague-Dawley (35)			0.3 mg/kg i.p. on day 1–3 1 mg/kg i.p. on day 4–7 3 mg/kg i.p. on day 8–11	↑ Ambulatory counts (OFT) ↓ Discrimination index (OLT)	[142]
♂ rat Sprague-Dawley (35)			2.5 mg/kg i.p. on day 1–3 5 mg/kg i.p. on day 4–7 10 mg/kg i.p. on day 8–11	↑ Exploration time (light–dark box test) ↓ Distance traveled (OF), exploration time (social motivation and social cognition test), % inhibitory prepulse (SR)	[117]
♂♀ rat Sprague-Dawley (35)			2.5 mg/kg i.p. on day 1–3 5 mg/kg i.p. on day 4–7 10 mg/kg i.p. on day 8–11	↑ Immobility time (FST) ↓ % of sucrose preference (SPT) = Open arm entries (EPM), time spent in center (OF), spontaneous locomotor activity	[119]
♂ rat Wistar (35)			2.5 mg/kg i.p. on day 1–3 5 mg/kg i.p. on day 4–7 10 mg/kg i.p. on day 8–11	↓ Time in open arms (EPM)	[121]
♂♀ rat Sprague-Dawley (32)			Self-administer escalating doses intravenously 3 μg/kg on day 1–3 10 μg/kg on day 4–6 30 μg/kg on day 7–20	↑ Discrimination index (delayed-match-to-sample working memory task)	[143]
♀ rat Sprague-Dawley (35)			2.5 mg/kg i.p. on day 1–3 5 mg/kg i.p. on day 4–7 10 mg/kg i.p. on day 8–11	↑ Immobility time (FST) ↓ Discrimination index (NOR), time spent in active social behaviors (SIT)	[138]
♂ mouse CD1 (41)	CBD	ECS stimulator	20 mg/kg i.p. 10 days	↑ Time spent in open arms (EPM), discrimination index (NORT)	[85]
♂ mouse Swiss albino (35)			3, 10, 30 mg/kg i.p. 15 days	↑ % open arm entries (EPM), latency for the first immobility episode (tail suspension test)	[83]
♂ rat Wistar (27)	WIN55,212-2	CB2 > CB1 agonist	1 mg/kg i.p. 20 days	↑ Latency to find the platform (MWM) ↓ Time in target area (MWM)	[93]
♂♀ rat (strain not specified) (45)			1.2 mg/kg i.p. 15 days	↓♂♀Anxiety index (OFT) ↓ ♂♀Impaired performance (OLT) ↓ ♂♀ Impaired performance on the social recognition test ↓ ♂ Impaired performance (NORT)	[130]
♀ rat Sprague-Dawley (30)			0.2 and 1.0 mg/kg i.p. 20 days	↑ Latency to feed (NSFT) ↓ Swimming and climbing (FST) = Time in open/close arms (EPM), distance traveled (OFT)	[124]
♂ rat Lewis (35)			2 mg/kg i.p. 13 days	↑ Latency to starle peak (SR), duration of exploration approaches (NORT) = Open arm entries, duration of open arm entries (EPM), social interaction	[92]
♂ rat Sprague-Dawley (35)	AM251	CB1 antagonist/inverse agonist	5 mg/kg i.p. 10 days	↓ Immobility duration (FST) = Time spent in open arm (EPM)	[131]
♂ rat Long-Evans (30)			1 mg/kg, i.p. 14 days	= Time in open arm (EPM), time of interaction (SIT)	[109]

Note: Increase (↑), decrease (↓), no change (=), male (♂), female (♀). Recurrent abbreviations listed in alphabetical order: Elevated plus maze (EPM), forced swim test (FST), Morris water maze (MWM), novel object recognition test (NORT), novelty suppressed feeding test (NSFT), object location task (OLT), open field test (OFT), social interaction test (SIT), startle reflex (SR), sucrose preference test (SPT).

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
