# Peer review of "Early Consumption of Cannabinoids: From Adult Neurogenesis to Behavior"

_ijms, 2021, doi:10.3390/ijms22147450_

Round 1

Reviewer 1 Report

The authors review the effects of early life cannabinoid exposure on brain development and behaviour.

There is a large amount of information here-and the manuscript would benefit from more critical insight. Instead of just listing the findings from studies, the authors should also make comments on the gaps in knowledge, limitations etc

For all of manuscript, the species should be clearly identified. This will affect the understanding of the information provided

Minor comments:

17 states in USA have legalised marijuana use (page 2)-this current number should be included as some makes it sound much less

"Researchers  have found no evidence of an effect on cannabis use or the perceived risk of use." this sentence should be expanded as it sounds like there is no risk for cannabis use-this is just the opinion of the study participants

Figure 1, though quite informative, is very busy. Can this be trimmed? Potentially a table with the inhibitors etc can be separated from the image itself?

Following Figure 1, the text is focused on ECS in the CNS. Recommend a subheading to clarify this

ECS development should be ECS during development

During development, (line 170+),  this is a rodent model. If so the authors should be explicit. Also acknowledging that development occurs postnatally in the rat for example until PN14

Line 218-various mammals-please name them

Line 259- , CBD dose seems to be a point of care to favor (3 mg/kg) or not favor (30 mg/kg) those aforementioned effects-please explain what this means?

With table 2, 3  why are only males reported? Are you indicating sex-specific differences? If so this needs to be explicit

Line 370-"The overall information discussed in this section "should be replaced with Thus research suggests

Line 372 under pathological conditions that affect the generation of new neuron-provide an example

Line 549- early and mid-adolescence-can you be more specific on the time? ie PN40-50?

Reviewer 2 Report

See attached file peer-review-12348083.v1.docx.
